# Vulnerability Assessment of Ecological–Economic–Social Systems in Urban Agglomerations in Arid Regions—A Case Study of Urumqi–Changji–Shihezi Urban Agglomeration

Xiaofen Zhang [1,2], Zibibula Simayi [1,2,*], Shengtian Yang [3], Yusuyunjiang Mamitimin [1,2], Fang Shen [1,2] and Yunyi Zhang [1,2]

1   College of Geography and Remote Sensing Sciences, Xinjiang University, Urumqi 830046, China
2   Key Laboratory of Wisdom City and Environment Modeling of Higher Education Institute, Urumqi 830046, China
3   Institute of Water Science, Beijing Normal University, Beijing 100875, China
*   Correspondence: zibibulla3283@sina.cn; Tel.: +86-13579267985

**Abstract:** This study aims to clarify the vulnerability characteristics of the ecological–economic–social system of oasis city clusters in arid zones, promote the deepening of research on the sustainable development of urban clusters, and provide crucial practical reference significance for solving the series of problems brought about by urbanization. This article takes the arid zone oasis city cluster, the Urumqi–Changji–Shihezi urban agglomeration, as the research object and constructs an indicator system from three dimensions of ecological environment, regional economic, and social development, and adopts the comprehensive index method, GeoDetector, the GM(1, 1) gray prediction model, and other methods to study the vulnerability pattern and spatial and temporal changes of the urban cluster from 2009 to 2018. The results show that (1) from 2009 to 2018, the change in the integrated ecological–economic–social system vulnerability index of the Urumqi–Changji–Shihezi urban agglomeration shows a general downward trend, followed by significant differences in the vulnerability of each dimension, with an average vulnerability index of 1.8846, 1.6377, and 0.9831 for the social vulnerability, regional economic, and ecological environment dimensions, respectively; (2) the evolution of the spatial pattern of changes in the vulnerability index of different systems in each region of the Urumqi–Changji–Shihezi urban agglomeration tends to change from large to slight spatial differences, in which the social and ecological environmental vulnerability changes are more prominent in addition to the vulnerability changes of the regional economy; (3) parkland area per capita, arable land area per capita, GDP per capita, social fixed asset investment, population density, and urban road area per capita are the main drivers of decreasing vulnerability of ecological–economic–social systems in urban agglomerations; (4) by predicting and calculating the vulnerability index of each region of the ecological–economic–social system of urban agglomerations, it is found that the vulnerability index of urban agglomerations will show a decreasing trend from 2009 to 2018, and the difference of the vulnerability index between systems will narrow; (5) finally, targeted countermeasures and suggestions to reduce the vulnerability of ecological–economic–social systems are proposed to provide scientific references for the sustainable development of arid oasis cities.

**Keywords:** vulnerability assessment; oasis urban agglomerations in arid zones; trend prediction; evaluation indicator system; Urumqi–Changji–Shihezi urban agglomeration

## 1. Introduction

The concept of vulnerability was first proposed by the scholar Timmernan P [1] in geography. With continuous research, vulnerability study gradually extended from natural disasters to other fields such as geography, ecology, economics, etc. Vulnerability-based research is maturing and has become one of the focal points of many international scientific institutions. The International Human Dimensions Program on Global Environmental

Change (IHDP) considers vulnerability one of its four core issues [2]. The definition of vulnerability varies depending on the object of study and disciplinary perspective, with the United Nations Intergovernmental Panel on Climate Change (IPCC) defining vulnerability in its Third Assessment Report as the degree of likelihood that a system, its subsystems, and system components are likely to cause damage under external stress [3]. Chuanglin Fang and other scholars [4] proposed that urban vulnerability refers to the coping ability of cities to resist disturbances from internal and external natural and human factors such as resources, ecological environment, and economic and social development in the development process. The city becomes vulnerable when this coping capacity against disturbances falls below a certain critical threshold. Vulnerability assessment, on the other hand, refers to the degree or likelihood of damage to a regional system adversely affected by certain anthropogenic activities through scientific methods [5].

Urban vulnerability assessment is increasingly becoming an essential issue on the policy agenda and in academia [6]. Research on urban vulnerability has grown considerably in the past few years but remains primarily limited by interdisciplinary differences in definition and scope [7]. The earliest studies of urban vulnerability focused on natural hazards and climate change, such as the vulnerability to natural hazards of specific cities or areas such as flood plains, coastal areas and seismic zones in specific external contexts; for example, C handra and Gaganis' study of the Nadi River Basin in the Fiji Islands found that the vulnerability to flooding in the basin was increasing [8]; Herslund et al. found a substantial increase in urban vulnerability due to climate change-induced risks in sub-Saharan African cities, and that continuing business-as-usual urban development patterns will reduce the resilience and adaptive capacity of cities to cope with the combined impacts of urbanization and climate change [9]; Tapia et al. introduced an indicator-based vulnerability assessment through five climate threats in 571 European cities which will facilitate the understanding of urban climate change risks and the development of effective adaptation policies [10]; and we note Kermanshah et al., Rasch, and Zhang et al., whose assessment of urban vulnerability to climate threats will help strengthen the adaptive capacity of cities in the face of climate change and natural disasters [11–13]. Urban vulnerability studies also focus on the vulnerability assessment of a city subsystem or the vulnerability of human–land coupled systems. The subsystems include the economic, ecological, environmental, and social systems. For example, some scholars, Rocha and Moreira, focus on the new market economy countries [14], and Ren Chongqiang and others use the Chinese provinces as the main study area and conduct a vulnerability analysis of the economic system in the study area [15]. In addition, Pan et al. conducted an ecosystem vulnerability analysis based on a habitat–structure–function framework in the Yangtze River basin in China [16]. In a study by Duy et al. in Vietnam, it was found that resilient transportation systems can reduce the vulnerability of cities to flooding [17], and Sterzel et al. assessed essential factors contributing to the significant differences in vulnerability through a study of rapid urbanization in coastal areas [18]. As times progress, there is a growing awareness of the cumulative impact of environmental, political, social, and economic risks on the ability of cities to function in times of shock and stress and a greater need to apply integrated research to understand the vulnerability of these rapidly growing cities to chronic and acute stresses and shocks [19]; for example, He et al. conducted a study on tourism-economy-ecosystem vulnerability in the Yangtze River Economic Zone to analyze the spatial and temporal evolution of its vulnerability and calculate the future vulnerability index prediction [20]; Chen et al. took Henan Province as an example to construct an evaluation model for vulnerability analysis from the coupled perspective of resource, ecological environment, economic and social vulnerability, and the results of the study showed that the overall vulnerability of Henan Province was decreasing between 2007 and 2016 [21]; moreover, scholar WU, R used Longnan city as the study area to measure the vulnerability of the coupled economic-social–ecological system of its districts and counties and concluded that the city should reduce vulnerability by enhancing the economic radiation capacity and improving the level of public services [22].

Urban vulnerability is also studied for specific types of cities. For example, assessing the vulnerability of resource-based cities is conducive to sustainable development [23–25]. Several scholars conducted a comprehensive and scientific analysis of the vulnerability characteristics of three Chinese cities with more than 10 million tons of coal mining. They identified the leading causes of natural and social vulnerability; the study is conducive to solving historical problems such as soil erosion and transforming the industrial structure to achieve sustainable urban economic development [26].

In natural disasters, climate change, economic, ecological, environmental, and social systems, and specific types of cities, many research results have been achieved in vulnerability assessment, and its technical methods have been initially developed. Some evaluation methods, such as the composite index method [27] and principal component analysis [28,29], have been widely used. The main components of current vulnerability assessment are quantitative evaluation models, such as the DRASTIC model [30], DPSIR model [31,32], FAHP model [33], etc., followed by the construction of indicator systems, such as the coastal vulnerability and social vulnerability indicator system [34], the ecosystem vulnerability assessment framework [35], the vulnerability framework of the coupled human-environment system [36], and the rural livelihood analysis framework [37]. There are other innovative technical approaches to vulnerability assessment; Hagenlocher, M et al. proposed an innovative approach based on a modular indicator library for assessing multi-hazard risks in global coastal deltas and internal social systems [38]; de Chazal, J. et al. proposed a methodology for assessing the vulnerability of social–ecological systems with direct correlation to multi-stakeholder values [39]. Metzger, M.J. et al. proposed a new ecosystem assessment methodology for ATEAM land use scenarios [40]; Teck, S. et al. evaluated marine ecosystem vulnerability using an expert evaluation method [41]; Thirumalaivasan, D. et al. developed an AHP-DRASTIC software package for specific aquifer vulnerability assessment studies [42]. Recently, GIS and remote sensing technologies have been combined with vulnerability assessment methods [43–47]. The vulnerability maps generated help reveal the spatial patterns of vulnerability and identify vulnerability hotspot areas.

The comprehensive study found that urban vulnerability studies are fragmented and relatively independent, leading to a lack of comparability between research data and making it challenging to assess by comparing data from independent studies [4]. Current urban vulnerability research is dominated by cities with high levels of economic development in the region. However, there is a paucity of research on cities with low levels of economic development in inland arid zones. Vulnerability studies in arid zones have mainly focused on single-system vulnerabilities such as economic systems [48,49] and ecosystems [50,51], while there are fewer studies on urban hybrid system vulnerability [52,53].

Under the background of an arid climate and natural environment, the ecological environment of urban clusters in the oasis zone is exceptionally fragile, and the rapid development of urban clusters and economic growth is exacerbating the pressure on the ecological environment; at the same time, under the unique urban development environment of the arid zone, the size of the oasis and the spatial distance between them restrict the economic activities of oasis cities to a certain extent. The oasis urban economy shows prominent vulnerability characteristics. Secondly, the fragile ecological environment and regional economy will inevitably pressure the social system. The fragility of the ecological environment, regional economy, and social system restrict urban clusters' upgrading and high-quality development. Studies based on the vulnerability of oasis cities in arid zones with low economic development levels are scarce, and the Urumqi–Changji–Shihezi urban agglomeration, as a new opportunity for the development of Xinjiang, is of strategic importance for the economic and social development of the whole Xinjiang and northwest regions.

Based on the above analysis, the main objectives of this paper are as follows: (1) to construct a research framework and a comprehensive evaluation system for the vulnerability of the ecological–economic–social system of the urban agglomeration; (2) to measure the

vulnerability of the ecological–economic–social tri-system of the Urumqi–Changji–Shihezi urban agglomeration from 2009 to 2018 and analyze its spatial and temporal evolution through the comprehensive index method; (3) to use the geographic detector to study the factors influencing the decline of the vulnerability index of the Urumqi–Changji–Shihezi urban agglomeration; (4) furthermore, use the gray prediction model to predict the vulnerability of the three systems of Urumqi–Changji–Shihezi urban agglomeration in the next seven years; (5) finally, to propose development measures for Urumqi–Changji–Shihezi urban agglomeration in response to the evaluation results to provide theoretical support for the sustainable development of the oasis urban agglomeration in the arid zone.

## 2. Materials and Methods

### 2.1. Study Area

The Urumqi–Changji–Shihezi urban agglomeration (Figure 1 below) is one of the 19 critical urban clusters planned for urban cluster construction at the national level and one of the eight typical representative urban clusters identified in the Urban System Planning of Xinjiang Uygur Autonomous Region (2012–2030). The total area of the Urumqi–Changji–Shihezi urban agglomeration is 63,800 km², with Urumqi as the "heart and brain", its scope also includes Shihezi, Changji, Fukang, Wujiaqu, Shawan, and Hutubi County, Manas County, and other six cities and two counties, as well as the sixth division, the eighth division, and the twelfth division of the Corps.

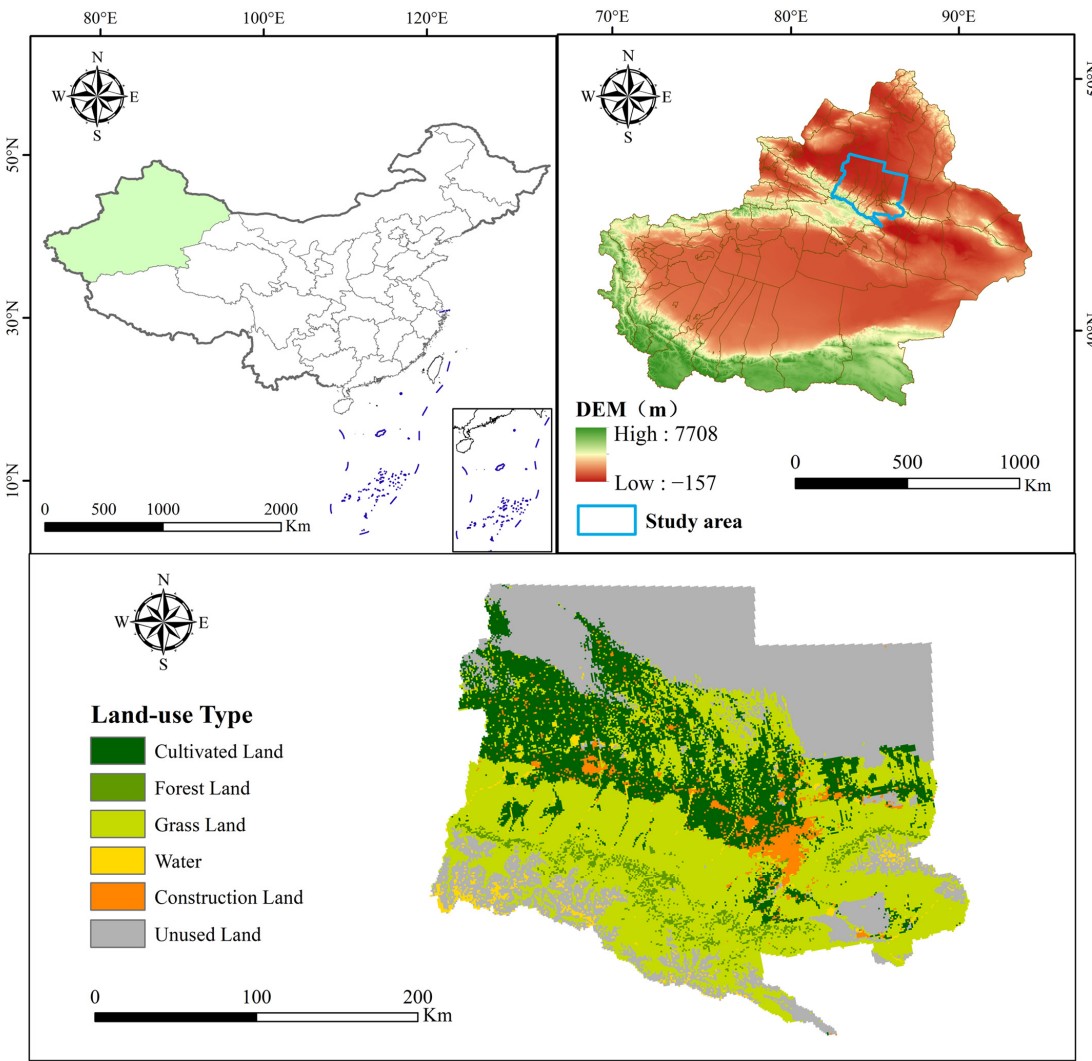

**Figure 1.** Overview of the Urumqi–Changji–Shihezi urban agglomeration.

Its natural conditions are superior; it is located on the southern edge of the Junggar Basin, with a moderate temperate climate and an annual average temperature of 5–7.5 °C; at the same time, the water and mineral resources are rich and there is an excellent economic base and convenient transportation; so, the carrying capacity of resources and environment is strong. The population and towns are also more concentrated; according to the data of the seventh census in 2020, the population is about 8.673 million people, accounting for more than 33.5% of the proportion of Xinjiang. With a gross regional product of 499.769 billion yuan, the Urumqi–Changji–Shihezi urban agglomeration is the future center and lifeline of economic and social development in Xinjiang. Due to the incomplete data for various urban development and national economic statistics of Wujiaqu City, Wujiaqu City was excluded as the study area in this paper for the vulnerability study of Urumqi–Changji–Shihezi urban agglomeration to ensure the scientificity and credibility of the evaluation results.

*2.2. Data Sources*

The data on various urban development and national economic statistical indicators involved in the study were mainly collected and compiled from the Xinjiang Statistical Yearbook, Urumqi Statistical Yearbook, Urumqi Yearbook, and the national economic and social development bulletins of counties and cities from 2010 to 2019. Some missing data were supplemented and perfected by interpolation.

*2.3. Research Indicator System*

This paper synthesizes the research on the three ecological–environment–regional economic–social development systems, follows the principles of completeness, uniqueness, objectivity, feasibility, and systematization, and has constructed a vulnerability evaluation index architecture ecological–economic–social system for the Urumqi–Changji–Shihezi agglomeration (Table 1 below). There are 23 evaluation indicators, which are divided into positive indicators and reverse indicators. The negative indicators are inversely proportional to the vulnerability of the system. The larger the value, the smaller the vulnerability; the positive indicators are positively proportional to the system's vulnerability. The larger the value, the greater the system's vulnerability.

**Table 1.** Ecological–economic–social vulnerability evaluation index system of Urumqi–Changji–Shihezi urban agglomeration.

| Target Layer | Criterion Layer | Index Layer | Indicator Description | Index Properties | Weight |
|---|---|---|---|---|---|
| Ecosystem Vulnerability | Ecological Vulnerability | Park green space per capita (m²-people) | Living environment and quality of life for urban residents | − | 0.017025 |
| | | Greening coverage of built-up areas (%) | Reflects the urban ecological environment | − | 0.0085627 |
| | | Cultivated land per capita (hectares) | Pressure on the ecosystem | + | 0.0091961 |
| | Environmental Vulnerability | Wastewater treatment rate (%) | Environmental Governance Capacity | − | 0.0086132 |
| | | Domestic waste removal volume (million tons) | Domestic waste treatment capacity | − | 0.0963664 |
| | | Total number of special vehicles for amenities and sanitation (units) | Environmental cleanliness protection capacity | − | 0.1175512 |
| Regional economic vulnerability | Economic structural vulnerability | The proportion of primary industry (%) | Reflect the level of regional modernization | + | 0.0188583 |
| | | Urbanization rate (%) | Reflects the urbanization process | − | 0.022392 |
| | | Share of industrial value added in GDP (%) | The pull of industry on the economy | − | 0.018683 |

**Table 1.** *Cont.*

| Target Layer | Criterion Layer | Index Layer | Indicator Description | Index Properties | Weight |
|---|---|---|---|---|---|
| | Economic efficiency vulnerability | Local revenue (billion yuan) | Reflects the degree of economic development | − | 0.1021245 |
| | | GDP per capita (RMB) | Economic level of regional residents per capita | − | 0.0142876 |
| | | Total retail sales of social consumer goods (million yuan) | Reflects the economic prosperity | − | 0.1009445 |
| | | Total social fixed asset investment (million yuan) | Reflects economic structure and quality | − | 0.0637112 |
| | Human Development Vulnerability | Population density (persons/km²) | Social Development Demographic Pressure Indicators | + | 0.0096287 |
| | | The average wage of employed workers (yuan) | Reflects regional wage levels | − | 0.0123461 |
| Social system vulnerability | Infrastructure Vulnerability | Urban road area per capita (m²) | Convenience of urban transportation | − | 0.0229304 |
| | | Drainage pipeline density (km/km²) | Reflects the city's sewage diversion capacity | − | 0.0236853 |
| | | Gas penetration rate (%) | Utility modernization level | − | 0.0070537 |
| | | Number of public toilets (one) | Sewage facilities construction capacity | − | 0.0965437 |
| | Social Environmental Vulnerability | Disposable income per urban resident (yuan) | Reflects the livelihood capacity and real standard of living of the society's residents | − | 0.0219353 |
| | | Net income per capita of rural residents (yuan) | | − | 0.0159044 |
| | | Number of beds in medical and health institutions (sheets) | City Public Service Levels | − | 0.0959529 |
| | | Number of urban basic pension insurance participants (persons) | Social Security Capability | − | 0.0957034 |

## 3. Research Methods

### 3.1. The Entropy Method

This paper uses the Entropy method to determine the weights of each evaluation index. The main steps are as follows.

(1) Standardization of the original data

According to the ecological–economic–social system evaluation system obtained in the previous section, to eliminate the influence of the different data outlines and size disparity on the different calculation results, the extreme difference standardization method is introduced to standardize the raw data.

For positive indicators, there are:

$$y_{ij} = \frac{x_{ij} - \min(x_{ij})}{\max(x_{ij}) - \min(x_{ij})} \tag{1}$$

For negative indicators, there are:

$$y_{ij} = \frac{\max(x_{ij}) - x_{ij}}{\max(x_{ij}) - \min(x_{ij})} \tag{2}$$

where $y_{ij}$ is the standardized data value; $x_{ij}$ is the original data value, where $i(i = 1, 2, 3, ..., m)$ is the number of sequences in the evaluation area; $j(j = 1, 2, 3, ..., n)$ is the number of evaluation index points; $\max(x_{ij})$ and $\min(x_{ij})$ are the maximum and minimum values of the $j$th index of the original data area $i$, respectively.

(2) Calculate the proportion of the $j$th indicator of region $i$ to this indicator

$$P_{ij} = \frac{x_{ij}}{\sum_{i=1}^{m} x_{ij}} \tag{3}$$

(3) Calculate the entropy value $E$ and information utility value $D$ of the $j$th indicator of region $i$

$$E_j = -(In\ m)^{-1} \sum_{i=1}^{m} P_{ij}, (j = 1, 2, ..., m); \tag{4}$$

$$D_j = 1 - E_j, (1 \ll j \ll n). \tag{5}$$

(4) Define the weight of the $j$th indicator

$$W_j = \frac{D_j}{\sum_{j=1}^{n} D_j} \tag{6}$$

where $W_j$ is the weight of the $j$th indicator of the $i$th evaluation object.

### 3.2. Vulnerability Evaluation Method

Vulnerability assessment is an essential element of current vulnerability research. In this paper, the composite index method, which is more commonly used to determine urban vulnerability, is used to evaluate the vulnerability of the ecological–economic–social system of the Urumqi–Changji–Shihezi urban agglomeration. The integrated index method establishes a system of evaluation indicators from the performance characteristics and causes of vulnerability. It uses statistical methods or other mathematical methods to synthesize them into a vulnerability index to express the relative magnitude of the vulnerability of the evaluation unit [5]. The vulnerability index is calculated by the following formula.

The vulnerability index for the second tier of indicators is calculated with the following formula.

Ecological system:

$$UVI_e = \sum_{i=1}^{m} y_{ij} \bullet W_{ij} \tag{7}$$

Regional Economic System:

$$UVI_f = \sum_{i=1}^{m} y_{ij} \bullet W_{ij} \tag{8}$$

Social System:

$$UVI_s = \sum_{i=1}^{m} y_{ij} \bullet W_{ij} \tag{9}$$

Ecological–economic–social complex systems:

$$UVI = UVI_e + UVI_f + UVI_s \tag{10}$$

where $W$ is the weight of each index, $UVI_e$, $UVI_f$, $UVI_s$ indicates the vulnerability index of each system of ecological environment, regional economy, and social development, respectively. The vulnerability of the ecological–economic–social complex system consists of the ecological and environmental system vulnerability $UVI_e$, the regional economic system vulnerability $UVI_f$, and the social development system vulnerability $UVI_s$ added together.

### 3.3. Vulnerability Classification Method

This paper classifies the vulnerability index of the Urumqi–Changji–Shihezi urban agglomeration based on the natural interruption point grading method in ArcGIS, with five vulnerability levels from low to high: Slight, Light, Medium, Heavy, Extreme, respectively (Table 2 below).

**Table 2.** Eco–environmental–regional economic–social system vulnerability level.

| Vulnerability Level | Slight | Light | Medium | Heavy | Extreme |
|---|---|---|---|---|---|
| Integrated system | ≤0.6301 | 0.6302~0.6670 | 0.6671~0.7647 | 0.7648~0.7756 | 0.7757~0.8387 |
| Ecological system | ≤0.1202 | 0.1203~0.1302 | 0.1303~0.1581 | 0.1582~0.1754 | 0.1755~0.2293 |
| Regional economic system | ≤0.2031 | 0.2032~0.2148 | 0.2149~0.2419 | 0.2420~0.2695 | 0.2696~0.3132 |
| Social system | 0.2941~0.2962 | 0.2963~0.3068 | 0.3069~0.3214 | 0.3215~0.3472 | 0.3473~0.3583 |

### 3.4. Geodetector

This paper uses GeoDetector to detect the main influencing factors of ecological–economic–social system vulnerability in the Urumqi–Changji–Shihezi urban agglomeration. GeoDetector is a set of statistical methods that are used to detect spatial differentiation and reveal the driving forces behind it [54], with the expression:

$$q = 1 - \frac{\sum_{h-1}^{L} N_h \sigma^2 h}{N \sigma^2} \qquad (11)$$

where $L$ is the stratification of ecological–economic–social system vulnerability $Y$ or indicator factor $X$ in the Urumqi–Changji–Shihezi urban agglomeration, $N_h$ and $\sigma^2 h$ are the number of cells and variance of layer $H$, respectively; $N$ and $\sigma^2$ represent the number of cells and variance of the study area, respectively. $q$ is the degree of influence of the indicator factor on the change in vulnerability, and a higher value of q indicates a stronger explanatory power of the indicator factor on vulnerability.

The factors x were classified into five categories by the natural interruption point hierarchy of ArcGIS, discretized, and the independent numerical variables were transformed into type quantities. Then the samples (Y, X) were read into the GeoDetector software to run the analysis, where the dependent variable Y was the vulnerability index. This paper investigates the degree of influence of each indicator factor of the ecological environment, regional economic, and social development systems on the integrated vulnerability of the urban agglomeration, respectively, during the decade. It focuses on the influence of each system's top two detection factors on the integrated vulnerability of the ecological–economic–social system of the Urumqi–Changji–Shihezi urban agglomeration.

### 3.5. GM(1, 1) Gray Prediction Model

In this paper, we use the gray prediction model to quantitatively predict the changes in vulnerability of the Urumqi–Changji–Shihezi urban agglomeration with the following steps and formulae [55]:

A. Let the time series $X_0 = \{x_0(1), x_0(2), \ldots, x_0(n)\}$ have $n$ observations and generate a new series $X_1 = \{x_1(1), x_1(2), \ldots, x_1(n)\}$ by accumulating the original series, then the corresponding differential equation of the GM(1, 1) model is

$$\frac{dX_1}{dt} + aX_1 = \mu \qquad (12)$$

where $a$ is the developmental ash number; $\mu$ is the endogenous control ash number;

B. Let $\hat{a}$ be the parameter vector to be estimated; which can be solved by using the least squares method to obtain and solve the differential equation to obtain the prediction model.

$$x_1{}^T \hat{X}_1(k+1) = [x_0(1) - \frac{\mu}{a}]e^{-ak} + \frac{\mu}{a}; \qquad (13)$$

$$(k = 1, 2, ..., n)$$

C. The accuracy test of the gray prediction formula is generally given in the following Table 3. If both P and C are within the allowed range, the predicted value of the indicator can be calculated. Otherwise, the formula needs to be re-corrected by analyzing the residual series.

**Table 3.** The accurary test grade of gray forecast model.

| Accuracy Class | P | C | Accuracy Class | P | C |
|---|---|---|---|---|---|
| High | >0.95 | <0.35 | Basic qualified | >0.70 | <0.65 |
| Qualified | >0.80 | <0.50 | Unqualified | ≤0.70 | ≥0.65 |

## 4. Results

### 4.1. Temporal Evolution Characteristics of the Combined Vulnerability of Urban Agglomerations

In this paper, based on the established ecological–economic–social system vulnerability index system of the Urumqi–Changji–Shihezi urban agglomeration, the weights of the evaluation indexes are determined using the entropy value method. The vulnerability indices of the composite and separate ecological–economic–social systems are calculated by the integrated index method. The calculated vulnerability indices are expressed visually using Origin 2021, as shown in Figure 2 below. Observing the changing trend of the vulnerability index of ecological–economic–social systems in the Urumqi–Changji–Shihezi urban agglomeration from a perspective of totality, it can be found that the overall change trend of the integrated vulnerability of the Urumqi–Changji–Shihezi urban agglomeration from 2009 to 2018 is slowly decreasing, and the vulnerability index of each system is decreasing in fluctuation. The composite vulnerability index decreases over the ten years, from 5.2653 to 3.9759. The composite vulnerability index decreases faster and then slower, using 2014 as the time point.

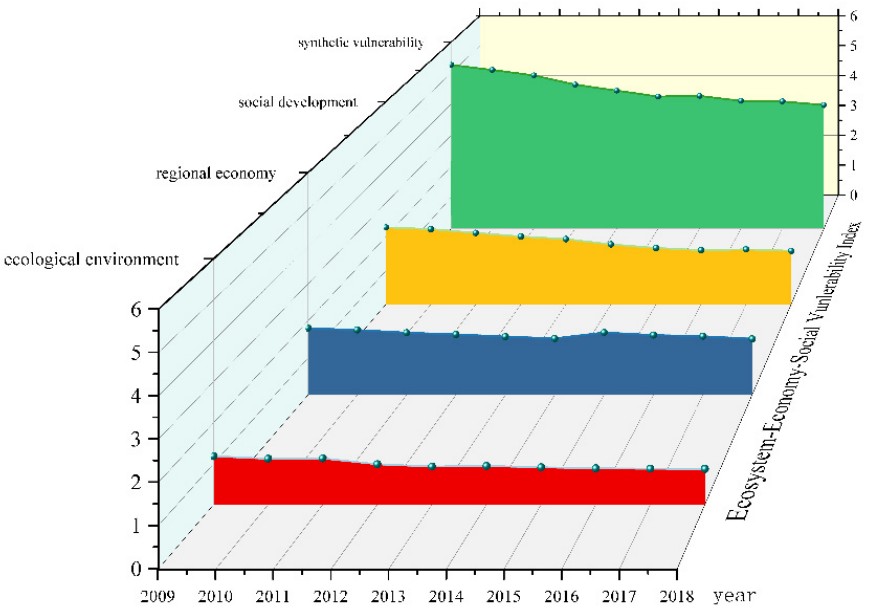

**Figure 2.** Trend of vulnerability index of ecological–economic–social system in Urumqi–Changji–Shihezi urban agglomeration.

### 4.2. Time Course of Subdimensional Vulnerability Evolution

The vulnerability of each dimension of the Urumqi–Changji–Shihezi urban agglomeration varies widely, with the highest social vulnerability, the second highest regional economic vulnerability, and the lowest ecological and environmental vulnerability, whose average vulnerability indices are 1.8846, 1.6377, and 0.9831, respectively. During the study period, the vulnerability of the social development system decreased significantly, the regional economic vulnerability showed a changing trend of decreasing, then increasing, and then decreasing, and the change of the vulnerability of the ecological environment system was more stable and showed a slow decreasing trend.

### 4.3. Spatial Differentiation Characteristics of Vulnerability of Subdimensional Urban Clusters

The vulnerability values of each region of the Urumqi–Changji–Shihezi urban agglomeration from 2009 to 2018 were spatially visualized using Arcgis 10.6 software and divided into three-time nodes—2009, 2013, and 2018—to obtain the spatial distribution of vulnerability levels in each region of the Urumqi–Changji–Shihezi urban agglomeration, as shown in Figure 3 below. From the figure, it can be found that the spatial pattern evolution trend of the vulnerability index changes of different systems in each region of the Urumqi–Changji–Shihezi urban agglomeration is as follows: overall, the difference of the spatial evolution pattern of the vulnerability index of ecological–economic–social systems in the Urumqi–Changji–Shihezi urban agglomeration changes from large to small, among which the change of regional economic vulnerability is smaller, and the change of social and ecological environmental vulnerability is larger.

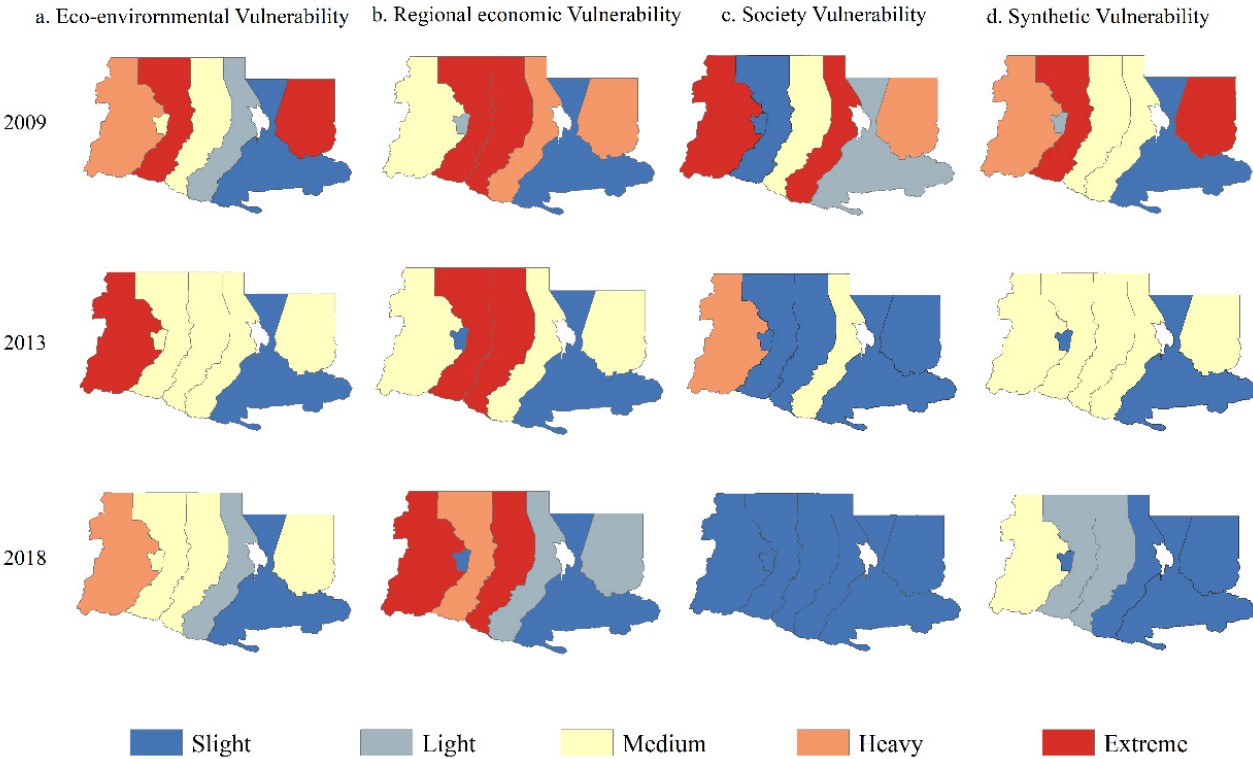

**Figure 3.** Spatial differentiation of sub-dimensions and integrated vulnerability of the Urumqi–Changji–Shihezi urban agglomeration.

Among the comprehensive vulnerability, the spatial pattern evolutionary representation of the vulnerability index in Urumqi is more stable and has been maintained at low vulnerability; the vulnerability level of the remaining areas, including Shawan, Shihezi, Changji, Fukang, Manas, and Hutubi counties, have all decreased, with Fukang having the most significant decrease.

The overall ecological vulnerability rank is higher in eastern cities and lower in western cities; the regional economic vulnerability rank has a considerable spatial variation. Among them, the spatial evolution pattern of vulnerability index levels in Urumqi, Hutubi County, and Shihezi City is very stable. It has been maintained at low, medium vulnerability, and medium, respectively. In contrast, the vulnerability levels in Manas County and Fukang City change significantly from high to medium vulnerability, while the vulnerability levels in Shawan City and Changji City rise and fall.

The spatial pattern of vulnerability levels in the regional economic dimension varies widely, with Manas County and Hutubi County in the central part of the region having high vulnerability levels in 2009, both of which have been high from 2009 to 2018, and the

vulnerability level of Shawan City has also increased from medium to high vulnerability. While Urumqi city has maintained a low vulnerability, the vulnerability levels of other regions, including Shihezi city, Manas county, Changji city, and Fukang city, have all decreased to different degrees.

The vulnerability level of the social dimension declined most significantly, with all of them gradually decreasing to low vulnerability from 2009 to 2018. Shawan and Changji cities were the most significant, gradually decreasing from high vulnerability to low vulnerability, while the vulnerability level of Shihezi city and Manas county was the most stable, remaining at low vulnerability. This is followed by Fukang city, Hutubi county, and Urumqi city, where the vulnerability levels have all decreased to different degrees.

### 4.4. Forecast of Ecological–Economic–Social System Vulnerability Development in the Urumqi–Changji–Shihezi Urban Agglomeration

As seen from Table 4, the vulnerability index of ecological–economic–social systems in the Urumqi–Changji–Shihezi urban agglomeration shows a decreasing trend, and the difference in vulnerability index between systems is reduced. Urumqi city has the lowest vulnerability index of 0.0721 in 2025, followed by Shihezi city, Changji city, Manas county, Fukang city, Hutubi county, and Shawan city, among which Shawan city is predicted to have the highest vulnerability index.

**Table 4.** Projected development of ecological–economic–social system vulnerability in the Urumqi–Changji–Shihezi urban agglomeration.

| Region | 2019 | 2020 | 2021 | 2022 | 2023 | 2024 | 2025 | 2019–2025 |
|---|---|---|---|---|---|---|---|---|
| Urumqi City | 0.2463 | 0.2154 | 0.1853 | 0.1559 | 0.1273 | 0.0993 | 0.0721 | 1.1016 |
| Shihezi City | 0.4853 | 0.4705 | 0.4562 | 0.4423 | 0.4288 | 0.4157 | 0.4030 | 3.1018 |
| Changji City | 0.5723 | 0.5565 | 0.5412 | 0.5263 | 0.5118 | 0.4977 | 0.4840 | 3.6898 |
| Fukang City | 0.5760 | 0.5607 | 0.5459 | 0.5314 | 0.5173 | 0.5036 | 0.4903 | 3.7252 |
| Hutubi County | 0.5911 | 0.5760 | 0.5614 | 0.5471 | 0.5331 | 0.5195 | 0.5063 | 3.8344 |
| Manas County | 0.5871 | 0.5660 | 0.5457 | 0.5261 | 0.5072 | 0.4889 | 0.4713 | 3.6923 |
| Shawan City | 0.6995 | 0.6929 | 0.6864 | 0.6799 | 0.6736 | 0.6672 | 0.6609 | 4.7605 |

## 5. Discussion

### 5.1. Dominant Factors Affecting the Vulnerability of Different Systems

5.1.1. Dominant Factors Affecting the Reduction of Ecosystem Vulnerability

As shown in Table 5, the average magnitude of the influence of the detection factors in the ecosystem on the changes in the vulnerability of the ecological–economic–social system in the Urumqi–Changji–Shihezi urban agglomeration is, in order, park green area per capita (X1) > arable land area per capita (X6) > the total number of vehicles dedicated to amenities and sanitation (X5) > green coverage of built-up areas (X2) > sewage treatment rate (X3) > domestic waste removal volume (X4). As an essential part of urban vegetation cover, urban green space systems can maintain urban ecosystem services and improve the human living environment [51], and insufficient vegetation cover is the main environmental factor [56] leading to land degradation. This is consistent with the findings of Penghua et al. and Zhang et al. [57,58]. The impact of arable land area per capita on ecosystem vulnerability in urban agglomerations is second only to parkland area per capita. Arable land is the type of land use on which humans depend and is an essential condition for ecosystem development. Therefore, the increase or decrease of arable land per capita impacts ecosystem vulnerability, consistent with Pan et al.'s findings [16]. The urban expansion causes land use change and thus decreases ecological vulnerability, especially in areas with significant land use change where agriculture and animal husbandry intermingle [59]. The oasis is mainly located in the north of Xinjiang. The economic development situation was good at the beginning of western development. The population is gradually dense, and the demand for agricultural land increases, so the per capita arable land area of the Urumqi–Changji–Shihezi urban agglomeration is rising. However, due to reasonable development and

utilization, there is no negative impact on the ecological environment, so the ecological environment vulnerability is reduced.

**Table 5.** Results of the GeoDetector of ecological–economic–social system vulnerability in the Urumqi–Changji–Shihezi urban agglomeration.

| Detection Factor | Detection Results by Years | | | | | | | | | |
|---|---|---|---|---|---|---|---|---|---|---|
| | 2009 | 2010 | 2011 | 2012 | 2013 | 2014 | 2015 | 2016 | 2017 | 2018 |
| X1 | 0.5943 | 0.8224 | 0.9488 | 0.8983 | 0.5713 | 0.8247 | 0.5997 | 0.6088 | 0.5482 | 0.5258 |
| X2 | 0.9080 | 0.9935 | 0.6632 | 0.4131 | 0.9974 | 0.3921 | 0.9058 | 0.6088 | 0.7807 | 0.9560 |
| X3 | 0.8388 | 0.9856 | 0.4889 | 0.9810 | 0.9953 | 0.9826 | 0.9479 | 0.9504 | 0.9305 | 0.3139 |
| X4 | 0.8483 | 0.9569 | 0.9728 | 0.9940 | 0.8553 | 0.9855 | 0.9013 | 0.9861 | 0.8419 | 0.8782 |
| X5 | 0.8874 | 0.8912 | 0.8872 | 0.9776 | 0.9801 | 0.9803 | 0.8575 | 0.9498 | 0.9412 | 0.9561 |
| X6 | 0.8700 | 0.9917 | 0.9519 | 0.9874 | 0.9883 | 0.9921 | 0.9565 | 0.9623 | 0.7802 | 0.7093 |
| X7 | 0.8700 | 0.9512 | 0.8886 | 0.9933 | 0.9987 | 0.8652 | 0.8323 | 0.8316 | 0.5250 | 0.4753 |
| X8 | 0.7780 | 0.8564 | 0.9107 | 0.8115 | 0.7107 | 0.7682 | 0.5164 | 0.5491 | 0.1648 | 0.4830 |
| X9 | 0.7254 | 0.8836 | 0.6210 | 0.9556 | 0.4482 | 0.9165 | 0.5643 | 0.5615 | 0.4149 | 0.9456 |
| X10 | 0.8144 | 0.8134 | 0.8872 | 0.9776 | 0.9816 | 0.9803 | 0.9437 | 0.9031 | 0.9579 | 0.9781 |
| X11 | 0.3709 | 0.9443 | 0.8259 | 0.9110 | 0.3792 | 0.3622 | 0.4532 | 0.4356 | 0.7808 | 0.7061 |
| X12 | 0.8884 | 0.7503 | 0.8799 | 0.9530 | 0.8506 | 0.9060 | 0.9408 | 0.9451 | 0.9551 | 0.9775 |
| X13 | 0.8458 | 0.8912 | 0.8930 | 0.9813 | 0.8506 | 0.9178 | 0.9368 | 0.9022 | 0.8999 | 0.9635 |
| X14 | 0.7790 | 0.5204 | 0.9519 | 0.9874 | 0.9953 | 0.9875 | 0.9566 | 0.9838 | 0.2513 | 0.3861 |
| X15 | 0.8483 | 0.9235 | 0.9525 | 0.4125 | 0.9987 | 0.4296 | 0.9928 | 0.5901 | 0.9158 | 0.9777 |
| X16 | 0.8270 | 0.8224 | 0.8723 | 0.8654 | 0.3457 | 0.8482 | 0.9058 | 0.9064 | 0.9478 | 0.9821 |
| X17 | 0.5972 | 0.5905 | 0.7297 | 0.8723 | 0.7825 | 0.9156 | 0.8134 | 0.7328 | 0.3613 | 0.5039 |
| X18 | 0.7800 | 0.9235 | 0.6423 | 0.5386 | 0.4035 | 0.8443 | 0.4478 | 0.6088 | 0.2198 | 0.2617 |
| X19 | 0.8874 | 0.8539 | 0.9599 | 0.9367 | 0.8382 | 0.9011 | 0.8675 | 0.8562 | 0.9277 | 0.9341 |
| X20 | 0.9382 | 0.5927 | 0.4233 | 0.2164 | 0.4529 | 0.2264 | 0.8386 | 0.8448 | 0.7577 | 0.9613 |
| X21 | 0.9226 | 0.9279 | 0.9519 | 0.9874 | 0.7825 | 0.8688 | 0.8134 | 0.8073 | 0.4790 | 0.2581 |
| X22 | 0.8874 | 0.7503 | 0.9273 | 0.9367 | 0.8449 | 0.9011 | 0.8675 | 0.8562 | 0.9692 | 0.9559 |
| X23 | 0.9450 | 0.9279 | 0.8886 | 0.9933 | 0.9987 | 0.9972 | 0.9119 | 0.9861 | 0.9692 | 0.9778 |

5.1.2. Dominant Factors Affecting the Vulnerability of Regional Economic Systems

The average magnitude of the influence of the detection factors in the regional economic system on the change of the vulnerability of the ecological–economic–social system in the Urumqi–Changji–Shihezi urban agglomeration is GDP per capita (X11) > the amount of social fixed asset investment (X13) > local fiscal revenue (X10) > urbanization rate (X8) > the proportion of industrial value added to GDP (X9) > total retail sales of social consumer goods (X12) > the proportion of primary industry (X17). GDP per capita, as an indicator reflecting the comprehensive strength of the economy, is the dominant factor in the regional economic system leading to the decreasing vulnerability of urban agglomerations, which is consistent with the findings of ChaoGAO et al., Liang and Xie, Lu et al., and Wang et al. [25,48,49,60]. GDP per capita is a complete manifestation of the economic capacity of supporting cities [61]. Its influence as a socioeconomic driver on the change of vulnerability of urban agglomerations is more active. The GDP per capita of the Urumqi–Changji–Shihezi urban agglomeration is on a growing trend. At the same time, the investment structure is continuously optimized. These factors are conducive to the stability of the economic system of urban agglomerations, making the urban economy more resilient [62], enhancing the risk resistance of the regional economic system, and reducing vulnerability. However, the weak economic foundation and the slow lag in regional economic development and industrial structure optimization will also restrict the degree of opening up of the regional economy to the outside world. Thus, GDP per capita will also hinder vulnerability reduction [63]. The total social fixed asset investment is an essential manifestation of economic structural vulnerability, which facilitates the effect on economic structural development [53] and has a significant positive effect on GDP. The rise of total social fixed asset investment can drive the economic improvement of the urban

agglomeration of the Urumqi–Changji–Shihezi while alleviating the vulnerability of the regional economic system of the urban agglomeration.

5.1.3. Dominant Factors Influencing the Development of Vulnerability in the Social System

The average magnitude of the influence of the detection factors in the social development system on the change of vulnerability of the ecological–economic–social system in the Urumqi–Changji–Shihezi urban agglomeration is population density (X14) > per capita urban road area (X16) > per capita net income of rural residents (X21) > number of beds in medical and health institutions (X22) > average wage of on-the-job workers (X15) > number of urban primary pension insurance participants (X23) > per capita disposable income of urban residents (X20) > the number of public toilets owned (X19) > density of drainage pipes (X17) > gas penetration rate (X18), in that order. Population density and urban road area per capita are the top two factors influencing the degree of vulnerability of social systems in the Urumqi–Changji–Shihezi urban agglomeration and are the key factors contributing to its declining social vulnerability. This is different from the results of YU et al. [64], and the main reason for the difference in results is the difference in evaluation indicators. The population has a more critical role in the vulnerability of the social system in Xinjiang. However, the rapid urbanization process leads to rapid population accumulation. The peak in population also increases the vulnerability of various aspects of urban development [53]. In contrast, the data collected and compiled on population density indicators show that the population density of the Urumqi–Changji–Shihezi urban agglomeration is decreasing, which can, to some extent, alleviate the pressure brought by the social system pressure. Inadequate infrastructure planning and construction is an essential factor affecting social vulnerability [21,65]. It is consistent with the results of LIU et al. [66] in the evaluation of social vulnerability in the Yellow River Delta region; per capita urban water supply, which reflects the degree of infrastructure support, is the main barrier factor. Urban road area per capita is an important indicator of urban accessibility, a sign of the increasing improvement of urban infrastructure, which is conducive to improving the coping capacity of urban social systems [67] and positively affects the reduction of the ecological–economic–social vulnerability of urban agglomerations.

*5.2. Policy Recommendations*

The overall vulnerability index of Urumqi–Changji–Shihezi urban agglomeration shows a decreasing trend, and the internal development process is more stable. Overall, the vulnerability index is lowest in Urumqi city on average, followed by Shihezi city, Changji city, Fukang city, Hutubi county, Manas county, and the highest urban vulnerability index in Shawan city. In the process of urban agglomeration development, different vulnerability risks are faced by different regions. Here, this paper proposes targeted measures to reduce the vulnerability index to make theoretical guidance for the sustainable development of the ecological, economic, and social systems of the Urumqi–Changji–Shihezi urban agglomeration.

Suggestions for the overall development of Urumqi–Changji–Shihezi urban agglomeration are as follows: the main problem of the current urban agglomeration is that the urban agglomeration is still in the primary development stage, with low integration in all aspects and unreasonable spatial structure. Given the main problems currently faced by the urban agglomeration, the development of the urban agglomeration should be promoted in the following directions in the future:

(1) Promote horizontal linkage among the cities of the Urumqi–Changji–Shihezi urban agglomeration and elevate its spatial structure optimization to a new level so that the overall economic strength of the urban agglomeration can be improved;

(2) Strengthen the core city driving role. Urumqi, as the core city of Urumqi–Changji–Shihezi urban agglomeration, will continuously improve its urban functions, play its economic radiation and driving role, and drive and lead the Urumqi–Changji–Shihezi urban agglomeration;

(3) Improve the level of industrialization of agriculture and animal husbandry and promote the modernization of agriculture and animal husbandry. The counties and cities in the city cluster of Urumqi–Changji–Shihezi that are dominated by agriculture and animal husbandry are mainly Manas County, Hutubi County, Shihezi City, Wujiaqu City, etc. They should actively cultivate characteristic advantageous industries such as cotton, animal husbandry, and agricultural products processing and promote the production base of raw materials for agricultural products processing and the integration of production, processing, and marketing operations;

(4) Coordinate to undertake industrial transfer and promote high-quality development;

(5) Improve the infrastructure network and strengthen the connection between inside and outside the city cluster. Not only should we pay attention to the construction of external transportation lines; we should also improve the intra-city clusters' transportation network, such as the one-hour transportation network of the Urumqi metropolitan area and the construction of Urumqi-Changji rail transit;

(6) Improve the quality of public services and jointly promote people's well-being. Urumqi–Changji–Shihezi urban agglomeration should promote the common construction and sharing of social security, education, medical care, etc., and enhance public safety and security capacity better to meet the full needs of the people's lives.

## 6. Conclusions

The perspective of this paper is ecological–economic–social system vulnerability, using the integrated index method and the gray prediction model to investigate the spatial-temporal evolution characteristics and future development trends of the three systems of the oasis city cluster in the arid zone—the Urumqi–Changji–Shihezi urban cluster—and explore the main influencing factors of ecological–economic–social system vulnerability of the Wuchang Shi urban cluster with the help of geographic probes. The main conclusions are as follows:

(1) Regarding time-series changes, the overall change trend of the comprehensive vulnerability of the Urumqi–Changji–Shihezi urban agglomeration during 2009–2018 is slowly decreasing, and the vulnerability index of each system is decreasing in fluctuation. Specifically, the ecological and environmental vulnerability decreased significantly during the study period, and the regional economic vulnerability showed a changing trend of first decreasing, then increasing, and then decreasing. The change in social system vulnerability was more stable and showed a slow decreasing trend;

(2) In terms of spatial evolution, the difference in the spatial evolution pattern of the vulnerability index of the ecological–economic–social system of the Urumqi–Changji–Shihezi urban agglomeration changes from large to small, where the magnitude of change in regional economic vulnerability is small, and the magnitude of change in social and ecological environmental vulnerability is large;

(3) In terms of influence factors, the parkland area per capita, arable land area per capita, and GDP per capita, the amount of fixed asset investment in the whole society, as well as the population density and urban road area per capita, are the top two influencing factors leading to the reduction of vulnerability in the ecosystem, economic system, and social system, respectively, and are also the key influencing factors for the reduction of vulnerability in the ecological–economic–social system of the Urumqi–Changji–Shihezi urban agglomeration;

(4) Regarding future projections, the vulnerability index of ecological–economic–social systems in the Urumqi–Changji–Shihezi urban agglomeration shows a decreasing trend, and the difference in vulnerability index between systems is reduced. The vulnerability index of Urumqi city is the lowest, with 0.0721 in 2025, followed by Shihezi city, Changji city, Manas county, Fukang city, Hutubi county, and Shawan city, among which Shawan county has the highest predicted vulnerability index.

**Author Contributions:** Conceptualization, X.Z. and Z.S.; methodology, X.Z.; software, X.Z.; validation, X.Z., F.S. and Y.Z.; formal analysis, X.Z.; data curation, X.Z.; writing—original draft preparation, X.Z.; writing—review and editing, X.Z.; project administration, Z.S. and Y.M.; funding acquisition, S.Y. All authors have read and agreed to the published version of the manuscript.

**Funding:** This research is funded by the National Natural Science Foundation of China (Grant No. 41661036), the National Natural Science Foundation of China—Xinjiang Joint Fund (Grant No. U1603241), and the Xin-jiang local government sent overseas study group supporting projects (No. 117/40299006).

**Institutional Review Board Statement:** Not applicable.

**Informed Consent Statement:** Not applicable.

**Data Availability Statement:** Data is contained within the article.

**Acknowledgments:** Thank you to the hard-working editors and reviewers for their valuable comments.

**Conflicts of Interest:** The authors declare that there are no conflict of interest.

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
