# Peer review of "Vulnerability Assessment of Ecological–Economic–Social Systems in Urban Agglomerations in Arid Regions—A Case Study of Urumqi–Changji–Shihezi Urban Agglomeration"

_sustainability, doi:10.3390/su15065414_

Round 1

Reviewer 1 Report

Vulnerability Assessment Study of Ecological-Economic-Social Systems in Oasis Urban Agglomerations in Arid Regions – Taking Urumqi-Changji-Shihezi Urban Agglomeration as an example

The paper assesses vulnerability in arid regions and try to solve the problems brought about by urbanization. The authors analysed in the detail many indicators.

However, several issues need to be addressed before the paper can be considered for publication:

·       it would be appropriate to add the main achieved results to the abstract

·       the discussions also contain recommendations for policy, which can be evaluated positively and contributes to the higher quality of the article and its importance for practice

·       in overall, the paper can be evaluated as interesting for real practice, and the results and recommendations are applicable

Reviewer 2 Report

The manuscript “Vulnerability Assessment Study of Ecological-Economic-Social Systems in Oasis Urban Agglomerations in Arid Regions” shows an assessment study of ecological-economic-social systems in the chosen region in China, and will be interesting for Sustainability readers, after revision.

Title: Vulnerability Assessment Study of Ecological-Economic-Social 2 Systems in Oasis Urban Agglomerations in Arid Regions---- Taking Urumqi-Changji-Shihezi Urban Agglomeration as an example, is too long, and should be improved by authors;

Aims: Abstract: As a weak point in the study (line 13) (…) – in Reviewer’s opinion will be better in this beginning: The aims/purpose is/are …;

Keywords: Vulnerability assessment; Oasis urban agglomerations in arid zones; Trend prediction; Evaluation indicator system; Urumqi-Changji-Shihezi Urban Agglomeration – please show this region shortly;

Vulnerability assessment – what does it mean? Please, specify this sentence;

The vulnerability index The Entropy method, China – will be better;

Materials and method, Discussion – well done;

Conclusion: Line 509-516 - please remove this text.

Literature: Chinese authors and researchers dominate this list, please look for authors from around the world (at least a few).

Overall, the research topic is interesting and well presented, the article is written correctly, and the considerations are presented understandably.
